# What We Know about and What Is New in Primary Aldosteronism

**DOI:** 10.3390/ijms25020900

**Published:** 2024-01-11

**Authors:** Natalia Ekman, Ashley B. Grossman, Dorota Dworakowska

**Affiliations:** 1Department of Hypertension & Diabetology, Medical University of Gdańsk, 80-214 Gdańsk, Poland; n.ekman@gumed.edu.pl; 2Centre for Endocrinology, Barts and the London School of Medicine and Dentistry, Queen Mary University of London, London E1 4NS, UK; ashley.grossman@ocdem.ox.ac.uk

**Keywords:** primary aldosteronism (PA), hypertension, cardiovascular risk in primary aldosteronism, adrenal vein sampling (AVS), radionuclide imaging, metomidate

## Abstract

Primary aldosteronism (PA), a significant and curable cause of secondary hypertension, is seen in 5–10% of hypertensive patients, with its prevalence contingent upon the severity of the hypertension. The principal aetiologies of PA include bilateral idiopathic hypertrophy (BIH) and aldosterone-producing adenomas (APAs), while the less frequent causes include unilateral hyperplasia, familial hyperaldosteronism (FH) types I-IV, aldosterone-producing carcinoma, and ectopic aldosterone synthesis. This condition, characterised by excessive aldosterone secretion, leads to augmented sodium and water reabsorption alongside potassium loss, culminating in distinct clinical hallmarks: elevated aldosterone levels, suppressed renin levels, and hypertension. Notably, hypokalaemia is present in only 28% of patients with PA and is not a primary indicator. The association of PA with an escalated cardiovascular risk profile, independent of blood pressure levels, is notable. Patients with PA exhibit a heightened incidence of cardiovascular events compared to counterparts with essential hypertension, matched for age, sex, and blood pressure levels. Despite its prevalence, PA remains frequently undiagnosed, underscoring the imperative for enhanced screening protocols. The diagnostic process for PA entails a tripartite assessment: the aldosterone/renin ratio (ARR) as the initial screening tool, followed by confirmatory and subtyping tests. A positive ARR necessitates confirmatory testing to rule out false positives. Subtyping, achieved through computed tomography and adrenal vein sampling, aims to distinguish between unilateral and bilateral PA forms, guiding targeted therapeutic strategies. New radionuclide imaging may facilitate and accelerate such subtyping and localisation. For unilateral adrenal adenoma or hyperplasia, surgical intervention is optimal, whereas bilateral idiopathic hyperplasia warrants treatment with mineralocorticoid antagonists (MRAs). This review amalgamates established and emerging insights into the management of primary aldosteronism.

## 1. Introduction

PA is the most common and curable cause of secondary hypertension [1]. PA was initially described by Lityński in Poland in 1953, then 2 years later in 1955 by Conn in the USA [2]. PA is associated with excessive and autonomous production of aldosterone. The three hallmark signs of PA are a high aldosterone blood level, a suppressed renin level, and hypertension [3]. Hypokalaemia had originally been considered a principal feature of PA; however, recent studies have reported that hypokalaemia is present only in 28% of PA patients. The most common causes of PA are bilateral idiopathic hypertrophy (BIH) (60% of cases) and aldosterone-producing adenomas (APAs; called Conn’s Syndrome) (35% of cases, although these percentages are changing over time) [2]. The rare causes of PA are unilateral hyperplasia, familial hyperaldosteronism (types I–IV), aldosterone-producing carcinoma, and ectopic aldosterone production [1]. PA was formerly thought to be a rare cause of mild-to-moderate hypertension (<1%) and hypokalaemia was one of the main conditions for diagnosis [4,5], but recent work has indicated that PA may be considered to be the most common cause of secondary hypertension [1]. PA is now considered to be a factor in 5–10% of all patients with hypertension [4,5,6]. Such studies have reported that the prevalence of PA depends on the severity of hypertension; the higher the degree of hypertension, the higher the prevalence of PA [7]. Thus, PA may occur in 20% of patients with resistant hypertension [8].

## 2. Physiology

Aldosterone is a steroid hormone produced in the adrenal zona glomerulosa. The main factors inducing the production of aldosterone are Angiotensin II and a high blood potassium level [4]. Other contributing factors, although to a lesser extent, include adrenocorticotropic hormone (ACTH), antidiuretic hormone (ADH), and β-endorphin. Aldosterone affects epithelial cells (the distal tube and collecting duct of the kidney, colonic mucosa, and sweat glands) through interaction with the mineralocorticoid receptor (MR). MRs are located in the cytoplasm of these cells. Aldosterone is lipophilic and easily diffuses through the cell membrane. Once there, aldosterone binds to MR, then the MR dimerises and translocates into the nucleus [1]. 

In the nucleus, dimerised MRs bind to a specific DNA-binding site and induce the process of gene transcription, leading to specific peptide products. In addition to its genomic function, aldosterone influences MR in a rapid non-genomic manner [5,9]. Aldosterone’s action on the distal tubes or collecting ducts of the kidney activates the production of the amiloride-sensitive epithelial sodium channel on the luminal side. Sodium is reabsorbed by this channel; the reabsorption of sodium impacts the hyperosmolar environment, causing the parallel reabsorption of water from urine to blood. This results in increased volume load [8]. Moreover, this process is accompanied by potassium excretion to obtain a balanced electrochemical gradient. Furthermore, aldosterone causes the urinary excretion of H+, causing metabolic alkalosis. Moreover, MR is also expressed in endothelial cells, vascular smooth muscle cells, cardiomyocytes, and some neurons [1]. Studies suggest that aldosterone binding to these receptors might cause oxidative stress, endothelial dysfunction, inflammation, and fibrosis in the walls of vessels or the heart [10]. Thus, hypertension of PA might be partly caused by sodium and water reabsorption and vasoconstriction of the arteries and may have direct cardiac effects [10].

## 3. Pathophysiology of Primary Aldosteronism

Primary aldosteronism is mainly caused by adrenal gland adenomas or bilateral idiopathic hypertrophy of the adrenal glands. In addition to these causes, there are also rare conditions related to PA including unilateral hypertrophy, familial hyperaldosteronism (types I–IV), aldosterone-producing carcinoma, and ectopic aldosterone production [11]. For many years, the precise pathogenetic mechanisms of these disorders were only poorly understood. Traditional aspects of PA diagnosis are mainly related to the differentiation between unilateral or bilateral forms of the disease, relevant to the appropriate therapy. 

In recent years, researchers have discovered many genetic abnormalities underlying sporadic and familial forms of PA and, as such, useful markers for improving diagnosis and treatment in the future.

Recent advances in understanding PA, especially concerning APAs, highlight the significant role of somatic mutations. Currently, the molecular characteristics of APAs are widely studied due to the development of molecular techniques and include such accurate analytic methods as exome sequencing, transcriptome sequencing, and the assessment of epigenetic changes such as DNA methylations in selected mutations [12]. Approximately 90% of APAs present somatic mutations that lead to aldosterone overproduction. These mutations are predominantly found in genes such as *KCNJ5, ATP2B3, ATP1A1, CACNA1D,* and *CACNA1H* [13].

The *CACNA1D* gene encodes a specific protein for a subunit of the L-type calcium channel Ca_V1.3. The main function of this channel is the transport of calcium ions into cells in response to depolarization. Mutation of the *CACNA1D* gene causes increased permeability to calcium ions, which finally induces aldosterone overproduction and cell proliferation in the adrenal zona glomerulosa. In addition, APAs with *CACNA1D* mutations are composed of zona glomerulosa-like cells and are smaller than APAs with *KCNJ5* mutations [14].

Other mutations that occur in APAs are *ATP1A1* and *ATP2A3*. *ATP1A1* and *ATP2B3* are genes that encode different types of ATPase enzymes; the *ATP1A1* gene encodes a subunit of Na+/K+ ATPase while *ATP2B3* encodes a plasma membrane Ca2+ ATPase. Mutations of these genes in APAs cause abnormal permeability to Na+ or H+ and increase aldosterone production. These mutations have not been observed in familial hyperaldosteronism [15].

One of the most common mutations reported in APAs is of *KCNJ5*, which encodes a potassium channel. Mutation of this gene causes increased sodium influx and subsequent depolarization, which leads to increased calcium levels in cells and finally stimulates aldosterone production. In addition, the presence of *KCNJ5* mutations in APAs is associated with certain clinical features such as more severe forms of hypertension, and more often occurs in female and younger patients [15]. In addition, the Japan Primary Aldosteronism Study (JPAS) conducted research on the molecular characteristics of *KCNJ5*-mutated APAs. This study demonstrated that APAs with somatic *KCNJ5* mutation have global DNA hypomethylation and transcriptomic profiles accompanied by changes in specific genes such as Wnt signalling, cytokine, and inflammatory response pathways. Thus, *KCNJ5*-mutated APAs constitute a specific subgroup of APAs [12]. Moreover, several studies investigating steroid profiling in PA diagnostics demonstrate that hybrid steroids 18-hydroxycortisol (*18OHF*) and 18-oxocortisol (*18oxoF*) are characteristic of APAs with *KCNJ5* mutations [16].

Somatic mutations present in the majority of APAs are generally responsible for the autonomous overproduction of aldosterone. Often the same mutations are discovered in patients with BIH and familial hyperaldosteronism (FH). There are several types of FH, each associated with different genetic mutations (germline mutations).

FH type I is characterized by severe hypertension in childhood with autosomal dominant inheritance. FH type I, also known as glucocorticoid-remediable aldosteronism, is caused by a unique genetic mutation involving the *CYP11B1* and *CYP11B2* genes. This mutation results in a chimeric gene combining elements of both genes; an unequal cross-over occurs between the *CYP11B1* gene, which encodes steroid 11β-hydroxylase, and the *CYP11B2* gene, responsible for aldosterone synthase. This chimeric gene produces aldosterone synthase, which is abnormally activated by ACTH instead of angiotensin II. These changes lead to the overproduction of aldosterone in patients with FH type I. However, this unique mutation also offers the possibility to control aldosterone production with glucocorticoids [17].

FH type II is associated with mutations in the *ClCN2* gene, which encodes the chloride channel ClC2. These changes cause the activation of calcium signalling and increased CYP11B2 expression on the membranes of zona glomerulosa cells. Diagnosis is based on clinical criteria and family history [18].

In FH type III, there are germline mutations in *KCNJ5*. This type of FH is often associated with severe hypertension. In addition, patients with FH type III can develop APAs and adrenal gland hyperplasia [19].

FH type IV is associated with mutations in *CACNA1H*, which encodes subunit calcium channels and activates calcium signalling. In each case, the result of these changes is inappropriate aldosterone biosynthesis [19].

In summary, most of the mutations discovered in PA patients are located in genes encoding ion channels and pumps. These mutations lead to increased calcium concentrations in the zona glomerulosa cells, which influences the expression of *CYP11B2* and, finally, the overproduction of aldosterone. Such new knowledge about genetic mutations related to the pathogenesis of PA could lead to the development of new pharmacological treatment targeting mutated proteins [20], whereas understanding the mutations that occur in FH could improve the diagnosis of positive patients in the early stages of the disease, before the onset of symptoms or cardiovascular complications [21]. In addition, the molecular characterisation of APAs using more advanced techniques, such as the analysis and integration of transcriptome and methylome, could provide new possibilities in the diagnosis and treatment of PA.

## 4. Primary Aldosteronism and Cardiovascular Risk

In recent years, numerous studies have reported that PA patients have a high rate of cardiovascular (CV) events [22,23,24,25]. Patients with PA more often experience myocardial infarction, heart failure, or stroke, and have a higher prevalence of atrial fibrillation (AF), independent of their blood pressure (BP) [22,24,26,27]. Compared to age-, sex- and degree of BP-matched patients with essential hypertension, patients with PA frequently show significant left ventricular hypertrophy (LVH0 [24,25] and increased aortic stiffness [23]. The pathogenesis is unclear and might be connected to excessive aldosterone blood concentrations, independent of BP, with a growing body of evidence suggesting that long-term exposure to high aldosterone levels might contribute to fibrosis and inflammation of the arterial wall and/or cardiac myocytes [10]. Aldosterone may also cause endothelial dysfunction by enhancing oxidative stress [10]. Thus, aldosterone excess has been associated with increased myocardial fibrosis, myocardial hypertrophy, and arterial stiffness [23]. A meta-analysis based on 12 case–control studies concluded that PA is significantly associated with sub-clinical arteriosclerosis and arterial stiffness [23] based on a comparison of CV markers including carotid intima-media thickness (CCA-IMT), flow-mediated dilation (FMD), nitroglycerine-mediated dilation (NMD), aortic pulse wave velocity (aortic-PWV), the augmentation index (ALX), the ankle-brachial index (ABI), and the prevalence of carotid plaques in patients specifically with PA. The researchers reported increased CCA-IMT and aortic-PWV in patients with PA. Moreover, compared with the normotensive control group, PA patients had increased ALX values and reduced FMD, which was related to increased aortic stiffness and impaired endothelial dysfunction [23]. According to data from the German Conn Syndrome Registry, cardiovascular mortality and morbidity are more common among PA patients than patients with comparable essential hypertension EH [28]. Nevertheless, overall mortality was not reported to be significantly different from a matched hypertensive control, although cardiovascular mortality was the main cause of death in PA (50% versus 34% in hypertensive controls) [28].

## 5. Signs and Symptoms

PA occurs mostly in patients between 20 and 60 years of age. Patients with PA have no pathognomonic symptoms. The three hallmark signs of PA are high aldosterone blood levels, a low/suppressed renin level, and hypertension. Hypertension in patients with PA is usually moderate to severe. A study from the Mayo Clinic in the USA found that the mean blood pressure in patients with PA was 184/112 ± 8/16 mmHg [3]. Other studies have reported that a higher prevalence of PA is seen in patients with higher elevations in BP [7]. Thus, PA was diagnosed in 20.9% of patients with ‘resistant hypertension’ [29]. In addition, patients with APAs have a higher aldosterone level and higher blood pressure than patients with BIH [3]. Hypokalaemia is recognised as a major feature of PA but occurs only in 28% of the total PA population [30]. The prevalence of hypokalaemia has been reported in 48% of APAs and 17% of BIH patients [7]. Thus, normokalaemic hypertension is currently the most common presentation of the disease.

Other symptoms associated with PA (Table 1) are headaches, visual problems, fatigue, muscle cramps, muscle weakness, numbness, and increased urination and thirst. Some of these symptoms are directly related to hypokalaemia. For example, polyuria is the result of a renal-concentrating defect caused by hypokalaemia. Symptoms and biochemical features of PA are listed in Table 1.

## 6. Screening of Primary Aldosteronism

For many years, PA has been thought to be a rare cause of hypertension, and screening was limited to patients with hypertension and hypokalaemia, considered as essential signs of PA. More recently, research has clearly shown that PA is the most common cause of secondary hypertension with a prevalence of 5–10% in hypertensive patients [3]. Hypertension in patients with PA is usually moderate to severe, and hypokalaemia occurs in 28% of PA patients [30]. Therefore, a hypertensive patient with normokalaemia should not be excluded from PA screening [32,33], and the degree of hypertension should be taken into account while screening for PA. Moreover, a particular group of young people (age < 35 years) have persistent and significant hypokalaemia but are not hypertensive. In these cases, APAs may be present. The absence of hypertension is probably the result of the early onset of the disease, its brief duration, and effective hypertension-preventing mechanisms [3]. Additionally, a review titled “How common is primary aldosteronism” emphasised that PA is highly prevalent within populations with mild-to-moderate hypertension, pre-hypertension, and even normotension [34]. Thus, screening for PA is challenging.

According to the Endocrine Society’s Clinical Practise Guidelines, it is recommended to consider PA in patients with:
Sustained blood pressure > 150/100 mmHg, confirmed on separate days.Resistant hypertension.Blood pressure < 140/90 mm Hg, requiring four or more anti-hypertensive medications.Hypertension with spontaneous or diuretic-induced hypokalaemia.Hypertension with an adrenal mass.Hypertension with sleep apnoea.Hypertension and a family history of early onset hypertension or stroke at a young age.Hypertension with a first-degree relative who has primary aldosteronism.

Irrespective of these factors, PA is known to be associated with higher cardiovascular morbidity and mortality than with essential hypertension [22] and is still very considerably underdiagnosed [35]. Currently, less than 1% of adults with diagnosed primary hypertension are screened for PA [6]. Early identification and specific treatment may reduce the risk of CV disease [33]. Thus, improvement in the extent of PA screening is required [36]. Increasing awareness and knowledge regarding PA among primary care physicians—who have the initial contact with the hypertensive patient—is crucial [6,37,38].

## 7. Diagnosis

Diagnosing of PA is complex and consists of three main assessments (Figure 1):
Case detection: ARR ratio assessment.Confirmatory testing: the oral sodium loading test, the saline infusion test, the fludrocortisone suppression test, or the captopril challenge test.Subtype classification: adrenal CT imaging and adrenal vein sampling.

## 8. Methods of ARR Measurement

According to the Endocrine Society Guidelines, the gold standard for the diagnosis of PA is the ARR. The ARR is the ratio of plasma aldosterone to plasma renin activity (PRA) or direct renin concentration (DRC). The circulating concentration of aldosterone is low compared with other steroid hormones; thus, the measurement of aldosterone is difficult and may be challenging. The most common method is radioimmunoassay, which was introduced some years ago. The priorities in the selection of measurement methods include the highest specificity and sensitivity. Therefore, high-performance liquid chromatography and tandem mass spectrometry measurement is currently considered the optimum assessment technique for the measurement of aldosterone (and indeed many other hormones) [39].

The measurement of renin levels is even more complicated because of two different possible markers—PRA and DRC [3]. PRA reflects enzymic activity, which is measured as the amount of angiotensin I generated from angiotensinogen over a period of time (in ng/mL/h). The DRC involves the direct measurement of the plasma renin concentration (usually in pg/mL). Some laboratories replace PRA with the more cost- and time-effective methods of DRC measurement; however, in some cases, PRA is a superior measurement to DRC. For example, oestrogen and progesterone affect the DRC (but not PRA), potentially causing false-positive screening for PA [37]. This is important for screening among women on oral contraceptive therapy [40] or hormone replacement therapy [41]. Thus, PRA assessment, where possible, is preferred [3].

Neither of these methods is perfect because accuracy is variable at low renin levels. Some authorities consider that the most promising solution to this issue is the measurement of angiotensin II as a direct factor regulating aldosterone synthesis [37]. Therefore, in the near future, the aldosterone/angiotensin ratio may be used to detect instances of PA.

## 9. ARR Testing

To increase sensitivity, ARR measurements should be performed following a certain protocol; aldosterone and renin samples should be collected in the morning after the patient has been out of bed for at least 2 h, and seated for 5 to 15 min directly before the blood sample collection [4,42]. The patient should also be prepared appropriately. Hypokalaemia should be corrected for a minimum of 24 h [33]. A few days before the test, the patient should discontinue any low-sodium diet, and mineralocorticoid antagonist (MRA) and other diuretics should be discontinued at least four weeks before testing [4].

The influence of medications on the renin–angiotensin system is important in the evaluation of the ARR. Many drugs (mainly anti-hypertensive drugs) may interfere with the ARR, causing false positive or false negative results (Table 2). Diuretics, angiotensin-converting enzyme inhibitors (ACEIs), angiotensin receptor blockers (ARBs), or dihydropyridine calcium channel blockers stimulating secretion of renin, may cause a false negative ARR. False positive ARRs have been reported among women on oral contraceptives or hormonal replacement therapy, but only when DRC was used to measurement of renin [37,40].

In most patients with severe hypertension, some anti-hypertensive drugs cannot be discontinued safely. In this case, the ARR should be determined during the evaluation of the results and potential interfering factors may be taken into account. Even when a patient cannot discontinue an MR antagonist, ARR testing can be performed as long as the renin is suppressed. In addition, there are also anti-hypertensive drugs that do not interfere with ARR. Among these medications are doxazosin, hydralazine, verapamil, prazosin, and terazosin; these drugs can be used during ARR testing [2].

It has been suggested that if the patient has an aldosterone level > 0.33 nM/dm^3^ and an ARR (with PRA) > 30 ng/mL/h or an ARR (with DRC) > 2.5 mU/L, the screening is positive. Sequentially, such patients should undergo a confirmatory test to exclude false positive results with the exception of patients with hypertension, spontaneous hypokalaemia, and an aldosterone level > 0.55 nM/dm^3^ and PRA < 1/ng/mL/h [4]. In these cases, the presentation of PA is diagnostic.

It should be noted that the specific diagnostic ARR is contingent upon the assay used, the units employed, and the assessment of renin and its detection limit. It is, therefore, in our opinion, essential that the clinician is aware of these diagnostic thresholds in the particular laboratory they use and should not simply employ a ratio taken arbitrarily from any publication.

In the SCOT-PA survey, researchers investigated the diversity of various aspects of diagnosing PA. This multinational, multicentre questionnaire-based study (33 centres) demonstrated significant diversity in the conditions of blood sampling, the assay methods of aldosterone and renin blood levels, and the methods and the cutoff points of screening and confirmatory tests. This heterogeneity of approaches to PA diagnosis was partly associated with a lack of specific guidelines, such as the lack of a gold standard for confirmatory testing. All these diagnostic differences may complicate the comparison of outcomes of PA patients across centres and also may have a significant impact on the diagnosis of patients with mild PA. The SCOT-PA survey has clearly shown that the standardisation of the PA diagnostic process is highly desirable [43].

## 10. Confirmatory Testing

A confirmatory test is the second step in the diagnosis of PA, following a positive result of ARR. In general, a confirmatory test is used to exclude false positive results of the screening tests. All recommended tests present higher negative predictive values than positive predictive values and are thus useful to exclude a diagnosis [44]. The four confirmatory tests (Table 3) are oral sodium loading, the saline infusion test (SIT), the fludrocortisone suppression test (FST), and the captopril challenge test (CCT). None of these tests is considered a gold standard, and thus the choice of a confirmatory test depends on patient compliance, cost, laboratory routine, or local preference and experience [33]. Some researchers consider the FST as the most reliable [45]; however, the FST is complex, time-consuming, and cumbersome. A recent prospective study reported that the SIT and CCT are reasonable alternatives to the FST [46]—these tests are easier to perform and more cost effective. We generally use the SIT as a simple and relatively inexpensive test, with the patient seated and taking care that the patient is initially normokalaemic and there is no concern regarding the fluid load.

Most patients who received a positive result of the ARR test undergo one of the confirmatory tests. However, for some groups of patients, who present with spontaneous hypokalaemia, plasma renin activity below detection levels, and PAC > 0.55 nM/dm^3^, a confirmatory test may not be necessary [4]. For these patients, the next recommended step is CT imaging.

## 11. Subtyping PA

The subtyping test currently involves computed tomography (CT) and adrenal vein sampling (AVS). The purpose of subtyping is to distinguish between unilateral and bilateral forms of PA. This is a crucial part of diagnosis, which is associated with the implementation of the most appropriate treatment. Unilateral adrenal adenomas or unilateral hyperplasia can be treated surgically; bilateral idiopathic hyperplasia should be treated with an MRA. Thus, the distinction is very important. Firstly, all patients with PA should undergo CT to exclude a large pathological mass in the adrenals, which could (very rarely) be a carcinoma [1,48]. Adrenal carcinomas are usually more than 4 cm in diameter and have a suspicious imaging phenotype on CT images [4]. APAs are small hypodense nodules on CT images and their size is usually less than 2 cm. BIH adrenal glands may be normal on CT images or may present thickening or nodular changes or show a non-functioning incidentaloma.

CT is the most sensitive imaging technique for identifying adrenal nodules [45]. CT is also more accurate than MRI imaging in this case. However, the sensitivity of CT is limited and not sufficient to detect small APAs nor distinguish between APA and a non-secreting incidentaloma [45,49].

A large number of studies have reported that the accuracy of CT when identifying APAs and differentiating them from other adenomas is poor [49,50,51]. In one study, PA patients underwent CT and AVS during the subtyping test. Based on this double evaluation, the researchers considered that CT was only 53% accurate [52]. This means that some patients might have had unnecessary and non-therapeutic surgery, while other patients could have been classified incorrectly and referred for pharmacological treatment instead of curative surgery. However, the “SPARTACUS” study reported contradictory conclusions: In this prospective randomised study, 184 patients were divided into two groups. One group received CT-based treatment and the other AVS-based treatment. At follow-up 1 year later, there were no significant differences between the groups in terms of the intensity of requisite anti-hypertensive medication [53]. Nevertheless, these conclusions are controversial; some researchers have stressed the inadequacies of the SPARTACUS trial, suggesting that it included an unrepresentative cohort, unusual end-points, and inappropriately low cutoff values for some laboratory results [3,53,54]. According to one authority, the main conclusion of this study is not that CT and AVS are equivalent for subtyping, but rather that a suboptimal AVS programme will have poor AVS-directed outcomes [3].

CT is an important and obligatory part of subtyping, and all PA patients should undergo CT imaging. However, the accuracy of CT is limited, and for patients who are surgical candidates, AVS is currently the accepted subsequent step in subtyping.

## 12. AVS

AVS is recommended as the principal subtyping test [3,42,55]. This method relies on the catheterization of both adrenal veins and the subsequent collection of blood samples from both adrenal veins and the inferior vena cava. AVS can be performed in three different ways, depending on the use of cosyntropin/*Synacthen* (a synthetic form of ACTH) for the stimulation of the adrenal glands during the procedure; without any stimulation, with a single injection of cosyntropin (250 µg, IV bolus), or with a continuous infusion of cosyntropin (50 ug/h, IV drip) [56]. Samples of cortisol and aldosterone are collected and subsequently assayed. Catheterisation of the right adrenal vein can be technically difficult [57]. Thus, the measurement of cortisol is used to evaluate successful catheterisation.

Successful catheterisation in AVS is generally determined by the selectivity index (SI), which is a measure of adrenal vein cortisol concentrations compared to peripheral vein cortisol levels. If all values above the cutoff points for SI confirm proper catheterisation [56], to distinguish between the unilateral and bilateral forms of the disease, the lateralisation index (LI) is evaluated. This index compares aldosterone-to-cortisol ratios between the left and right adrenal veins. In recent years, various guidelines have reported different cutoff points for the selective index and lateralisation index. According to the European consensus of 2020, a selectivity index ≥ 2 for unstimulated AVS and SI ≥ 5 for stimulated AVS is considered indicative of successful catheterisation, confirming that the sample was indeed taken from the adrenal vein [58]. A lateralisation index greater than 4.0 is indicative of a unilateral form of the disease. In contrast, a lower value suggests bilateral aldosterone overproduction.

AVS is a technically demanding procedure and requires an experienced interventional radiologist [50]. This approach should be performed in referral centres to optimise successful catheterisation of all relevant sites and minimise the risk of complications such as groin haematoma, thrombosis, adrenal haemorrhage, or dissection of an adrenal vein [50]; there have been reported complication rates of <2.5 in AVS-experienced centres [52].

According to current guidelines, AVS should be offered to all patients with a confirmed diagnosis of PA who can and wish to undergo surgical treatment [4,49]. However, it has been contended that AVS is not necessary for young patients under 35 years of age with a confirmed PA (a high ARR, undetectable renin, and spontaneous hypokalaemia) and radiological features of unilateral APA with a contralateral normal adrenal. This group of patients should be offered adrenalectomy directly.

Researchers from the JPAS (Japan Primary Aldosteronism Study) conducted a study investigating a combination of CT findings and serum potassium levels in subtype diagnosis based on AVS. In this study, 1591 patients with PA were included and, before AVS, they underwent CT scans, and serum potassium levels were evaluated. The results of this study showed that patients with bilateral normal results based on CT and normokalaemia could be treated pharmacologically because the probability of unilateral disease was low. In contrast, patients with unilateral abnormalities in CT scans and hypokalaemia were associated with a high probability of unilateral disease based on AVS. Thus, AVS was strongly indicated for this group of patients [59].

## 13. Non-Invasive Alternatives to AVS

AVS is a recommended method used to distinguish between the unilateral and bilateral forms of PA. This method presents high sensitivity and specificity and can guide surgical decisions effectively [60]; however, it has some disadvantages too. This procedure is technically demanding, requires an experienced interventional radiologist, and is invasive, which can cause potential complications such as groin haematomas, adrenal vein thrombosis, vein perforation, and even adrenal haematomas [61].

Studies have, therefore, been carried out to explore non-invasive methods as alternatives to AVS. One of these methods is ^11^C-metomidate PET (positron emission tomography) [3]. Metomidate is a selective inhibitor of the adrenal steroidogenic enzyme 11β-hydroxylase (CYP11B2) and shows high specificity for adrenal gland tissue containing this enzyme. ^11^C-metomidate binding is correlated with this enzyme activity and can be potentially useful as a PET radiotracer for PA diagnosis [62]. Over the past few years, several prospective studies have been conducted to compare ^11^C-MTO-PET (^11^C metomidate PET) to AVS to detect lateralisation in PA. A study by Soinio and colleagues in 2020 investigated the lateralisation accuracy of ^11^C-MTO-PET in PA patients compared with AVS lateralisation and evaluated outcomes after adrenalectomy. This study included 55 patients with confirmed PA (3 patients were excluded) who underwent AVS and ^11^C-MTO-PET imaging in random order. ^11^C-MTO-PET concordance with AVS in the adrenalectomy group was 53% among patients with CYP11B2-positive APA and 60% among patients with non-APA [63]. In another prospective study of 143 PA patients, the accuracy of ^11^C-MTO-PET was compared with AVS when predicting biochemical and clinical remission after adrenalectomy: ^11^C-MTO-PET imaging was superior to AVS in predicting remission of PA after surgery, although this was not statistically significant. ^11^-C-MTO-PET predicted biochemical remission in 72.7% of patients, with the resolution of hypertension in 65.4%, while for AVS, the respective percentages were 63.6% and 61.5% [64].

These studies suggest that ^11^C-MTO-PET is a promising approach and has potential as a non-invasive method for PA diagnosis and detecting adrenocortical masses. However, further research is necessary, especially in comparison to current standard methods such as AVS. In addition, ^11^C requires a local cyclotron, such that other more easily available isotopes such as ^18^F are currently under investigation.

## 14. Surgical Treatment

Surgery is the optimal method of treatment for patients with APA or unilateral adrenal hyperplasia [65]. Before an operation, the patient should be prepared appropriately. Blood pressure should be controlled with anti-hypertensive drugs consisting of MRA, and hypokalaemia should be corrected. Preoperative MRA applied a few weeks before surgery may decrease the risk of post-operative hypoaldosteronism associated with chronic suppression in the contralateral gland [65].

The ideal method of surgery is laparoscopic adrenalectomy. An alternative to laparoscopy is represented by open transperitoneal adrenalectomy. Compared with open surgery, laparoscopy is safer and associated with fewer complications, shorter hospitalisation, and smaller incisions [66]. There are two types of laparoscopic adrenalectomy—lateral transperitoneal adrenalectomy (LTA) and laparoscopic posterior retroperitoneal adrenalectomy (LPRA). LTA is the most common and widely used method of surgery [67]. However, recent studies have reported some advantages of LTRA compared with LTA; LPRA involves direct access to the adrenal gland, no manipulation of peritoneal organs, shorter operation times, less blood loss, less post-operative pain, and a shorter length of hospitalisation [66,67,68]. Regardless of the method of surgery, the entire adrenal gland is removed during the operation. Generally, total adrenalectomy is recommended because APAs are small and difficult for the surgeon to identify [69]. However, some studies comparing partial adrenalectomy versus total adrenalectomy have reported similar therapeutic outcomes [70,71].

Post-operatively, there is a risk of temporary hypoaldosteronism, which is associated with suppression of the contralateral adrenal gland [72]. This may cause clinically relevant hyperkalaemia. Thus, to avoid hyperkalaemia, MRA and potassium supplements should be discontinued directly after surgery, and serum potassium levels should be monitored weekly for 4 weeks [3]. The total dosage of anti-hypertensive drugs should be reduced by 50%. Shortly after surgery, aldosterone blood levels and renin activity should be measured as an early indication of a biochemical response [65].

Moreover, some researchers have reported that older patients and patients with long-lasting PA may demonstrate deterioration of renal function and develop transient or persistent insufficiency of the zona glomerulosa after unilateral adrenalectomy [73]. This may require fludrocortisone supplementation. Other studies have shown that younger patients and female patients have a higher chance of complete clinical success after adrenalectomy [74]. Elevated blood pressure is controlled without or with a reduced dose of anti-hypertensive drugs, and the cardiovascular risk is significantly decreased [33].

It should also be noted that a significant proportion of patients may co-secrete cortisol, leading to contralateral adrenal cortisol suppression and a period of cortisol deficiency. This should be monitored and consideration given to transient hydrocortisone replacement [75].

Finally, there has been a recent move to consider left adrenalectomy via endoscopic access through the stomach, but the advantages and pitfalls of this technique remain to be assessed.

In addition, researchers are constantly investigating new, less-invasive methods of surgical treatment for patients with unilateral PA. One of these methods could be adrenal thermoablation as a treatment option for APAs in patients with PA. Thermoablation involves the use of temperatures above 50 °C to selectively remove the entire tumour without disturbing the surrounding healthy tissue. The advantage of thermoablation over adrenalectomy is the ability to spare the surrounding normal adrenal tissue and, thus, reduce the risk of adrenal insufficiency [76].

The article “Aldosterone-producing Adenoma: Considerations on in Vitro Effects of Adrenal Thermoablation and its in Vivo Applications” suggests that thermal ablation can be effective in controlling blood pressure and reducing pharmacotherapy. The use of thermal ablation may provide a significant reduction in aldosterone levels and blood pressure in patients with APA [76].

In summary, thermal ablation is a promising treatment option for APAs but some disadvantages and limitations should be considered, such as lifelong adrenocortical insufficiency, appropriate technological design, advanced treatment planning, and a lack of a standardised protocol and relevant guidelines [77]. Therefore, further research and clinical trials on the long-term safety and effectiveness of thermoablation in APA treatment are needed.

## 15. Pharmacological Treatment

Pharmacological treatment of PA is indicated for patients with BIH and also for patients with the unilateral form of the disease who cannot undergo surgery [1]. However, it should be noted that control and survival are significantly better with surgery as opposed to medical therapy [78]. Pharmacological treatment is based on mineralocorticoid receptor antagonists. Currently, only two drugs of this type are available—spironolactone and eplerenone [4].

Spironolactone is the first generation of MRA that was approved by the FDA in the 1960s and has been an essential drug in the treatment of PA [3]. It is inexpensive and widely available. The starting dosage of spironolactone is 12.5–25 mg per day in a single dose. Spironolactone has a slow onset of action and, therefore, the dose should be monitored every 4 weeks [1]. The aim of treatment is effective mineralocorticoid receptor blockade, which is aimed to achieve blood pressure control, potassium level normalization, and the prevention of excess aldosterone-associated organ damage [2]. Spironolactone should be titrated to a high–normal level of potassium without oral potassium supplements [3]. The maximum dosage of spironolactone is 400 mg per day.

However, spironolactone is a non-selective MRA. This means that it influences not only MRs but also other receptors. Spironolactone is the antagonist of testosterone receptors and the agonist of progesterone receptors [79]. Thus, during treatment, endocrine side effects may occur: The anti-androgen effects of spironolactone may cause gynaecomastia, erectile dysfunction, and decreased libido in men. In women, the agonist activity at the progesterone receptors may induce menstrual irregularities. The side effects of spironolactone are dependent on dosage [57]. One study has reported that gynaecomastia occurred in 6.9% of men who received < 50 mg/d and in 52% of men who received > 150 mg/d [80]. Therefore, to avoid or decrease endocrine side effects, amiloride or a small dose of thiazide diuretic can be added to lower the dose of spironolactone [4]. Generally, administration of spironolactone in men should be up to 50 mg/day, but if higher doses are necessary an alternative would be considered to avoid causing sexual dysfunction.

A second-generation MRA is eplerenone, which was approved by the FDA in 2003. Compared with spironolactone, eplerenone is more selective, and endocrine side effects occur rarely [79]. The initial dose of eplerenone is 50 mg daily. Eplerenone has a shorter half-life and must be administrated twice a day. Despite better tolerability, eplerenone is more expensive and probably less effective than spironolactone. One double-blind study reported that spironolactone is superior to eplerenone at decreasing BP in patients with PA [81], but the dose of spironolactone was higher than usually applied and eplerenone was administrated only once a day.

Currently, researchers have been working on third- and fourth-generation MRAs. Representative third-generation MRAs should be non-steroidal with high specificity for MR (higher than eplerenone), cheap, and a long half-life. Fourth-generation MRAs should, in addition, be tubule-sparing to decrease the risk of hyperkalaemia [6].

In addition, some studies have reported that MRAs such as spironolactone have inhibitory effects on steroidogenesis and aldosterone production. This effect might contribute to the spontaneous remission of PA after long-term treatment with MRAs in some patients. However, more research is needed to identify the specific patient characteristics, better understand underlying mechanisms, and determine the optimal treatment durations for MRAs [82].

## Figures and Tables

**Figure 1 ijms-25-00900-f001:**
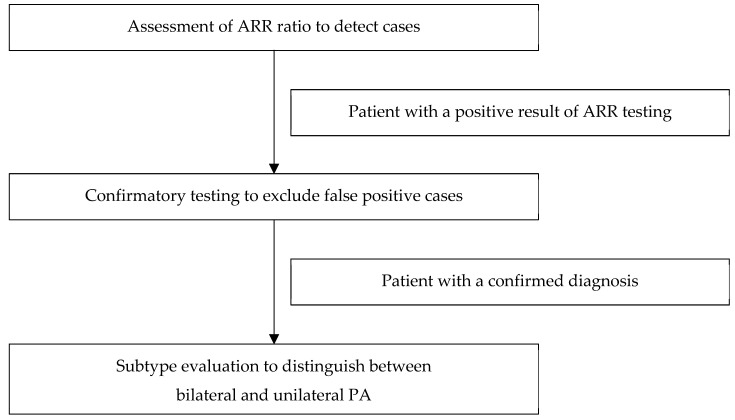
Model of the diagnostic pathway of primary aldosteronism.

**Table 1 ijms-25-00900-t001:** Symptoms and associated biochemical changes seen in PA [31].

Symptoms	Biochemical Features
Hypertension	
Headaches	High aldosterone level
Visual problems	Low renin level
Fatigue	Hypokalaemia
Muscle cramps and weakness	Hypernatraemia
Numbness	Metabolic alkalosis
Increased thirst and polyuria	

**Table 2 ijms-25-00900-t002:** The potential confounding influence of drugs on ARR [37].

Name of Drug	Potential Confounding Influence of Drug on ARR
Diuretics	Reduce
ACE inhibitors/ARBs	Reduce
β-adrenoceptor blockers	Increase
SSRI	Reduce
Hormone replacement therapy	Increase (if DRC is used)
Contraceptive therapy	Increase (if DRC is used)

**Table 3 ijms-25-00900-t003:** Confirmatory tests used in the second step of PA diagnosis [47].

Type of Confirmatory Tests	Oral Sodium Loading	Intravenous Saline Infusion	Fludrocortisone Test	Captopril Test
**Procedure**	- For 3 days patient should increase sodium intake > 200 mmol/d (6 g/d) - In the morning of the third day of high sodium diet, patients start 24 h urine collection - After 24 h urine collection, aldosterone and sodium is measured - During test potassium blood level should be controlled daily	- Patients stay in the recumbent position for at least 1 h before beginning the test - then patients receive a 2 L 0.9% NaCl infusion intravenously for 4 h and remain in a seated position - Aldosterone blood level and renin blood level is measured at the beginning of test and after 4 h - Before the test, hypokalaemia should by corrected	- For 4 days, patients receive 0.1 mg of fludrocortisone orally every 6 h together with KCl supplements (to keep plasma K+ close to 4.0 mmol/L) and NaCl supplements to maintain a urinary sodium excretion rate of at least 3 mmol/kg body weight. - On day 4, aldosterone and renin blood levels are measured at 10 A.M. with the patient in a seated posture, and cortisol blood level is measured at 7 and 10 A.M.	- Patients are sitting or standing at least 1 h before beginning the test - Patients then receive 25 mg of captopril orally - Measurements of aldosterone, renin, and cortisol blood levels are made at the beginning of test and 1 h and 2 h after captopril - The patient is sitting during the test
**Interpretation**	- Diagnosis of PA is positive when urine aldosterone secretion is > 12 ug/d - Sodium is measured in urine to control if high sodium diet was content	- Diagnosis of PA is positive when post infusion aldosterone blood > 0.28 nM/dm^3^ - A result of < 5 ng/dL is negative, but outcomes of 0.14–0.28 nM/dm^3^ and 5–10 ng/dL are questionable	- Diagnosis of PA is positive when, on day 4 at 10 A.M., the aldosterone blood level is > 0.17 nM/dm^3^ 6 ng/dL, the renin blood level is suppressed (PRA < 1 ng/mL/h), and plasma the cortisol blood level is lower at 10 A.M. than at 7 A.M. (to exclude a confounding ACTH effect)	- Diagnosis of PA is positive when the aldosterone blood level is not suppressed after captopril and the renin blood level remains suppressed - Normally, the aldosterone blood level is suppressed by captopril (> 30%)
**Comments**	This test is not recommended by patients with severe uncontrolled hypertension, severe hypokalaemia, arrhythmia, and renal insufficiency	This test is not recommended by patients with severe uncontrolled hypertension, severe hypokalaemia, arrhythmia or renal insufficiency	Some specialists consider, that FST is the most sensitive test for confirming PA and it is a less intrusive method of sodium loading	Occasionally, in patients with BIH, a decrease in aldosterone levels is seen as a result of some false negative or inconclusive outcome

## Data Availability

Not applicable.

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
