# Peer review of "What We Know about and What Is New in Primary Aldosteronism"

_ijms, 2024, doi:10.3390/ijms25020900_

Round 1

Reviewer 1 Report

Comments and Suggestions for Authors

This is interesting and well-written review. I have just few comments.

Major:

From line 292, authors stated about variability of the measurements of aldosterone. Including screening test and confirmatory test, the diversity of approaches utilized for each diagnostic step in different expert centers were investigated in SCOT-PA (PMID: 36726325). I recommend authors cite this article and discuss this problem.

From Line 382, the recommendation for AVS is mentioned. The Japanese JPAS study (PMID: 29092077), which determines the AVS recommendation based on CT finding and potassium values, may be helpful for discussion.

The pathogenesis of PA has been discussed since Line 79, focusing on somatic mutations. Molecular biological analysis focusing on somatic mutations is also in progress, including studies focusing on the transcriptome and methylome of KCNJ5 somatic mutations (e.g. PMID: 28747387) and on the production of hybrid steroid (e.g. PMID: 32206594, PMID: 31484828), which might be cited and mentioned as appropriate.

Minor:

Line 45: Line 45: What is meant by the description "FN"? Is it not FH (familial hyperaldosteronism)?

Line 83: There is a description of ectopic aldosterone production. It is considered to be a very rare condition. Do authors have any references?

Line 384:”AVA” should be “AVS”.

Line 389: “ASV” should be “AVS”.

Author Response

Dear Reviewer,

Thank you for your insightful review and the impact it had on our manuscript. We appreciate the time and effort you dedicated to reviewing our work.

The answers to your comments can be found below.   Kind regards,   Natalia Ekman

Q1: From line 292, authors stated about variability of the measurements of aldosterone. Including screening test and confirmatory test, the diversity of approaches utilized for each diagnostic step in different expert centers were investigated in SCOT-PA (PMID: 36726325). I recommend authors cite this article and discuss this problem.

Answers to Q1: We were following your suggestion, and we added a discussion of this problem and appropriate references in our manuscript:

„In the SCOT-PA survey researchers investigated the diversity in various aspects of diagnosing PA. This multinational, multicentre questionnaire-based study (33 centres) demonstrated significant diversity in the conditions of blood sampling, assay methods of aldosterone and renin blood levels, and the methods and the cutoff points of screening and confirmatory tests. This heterogeneity of approaches to PA diagnosis was partly associated with a lack of specific guidelines, such as the lack of a gold standard for confirmatory testing. All these diagnostic differences may complicate the comparison of outcomes of PA patients across centres and also may have a significant impact on the diagnosis of patients with mild PA. The SCOT-PA has clearly shown that standardization of the PA diagnostic process is highly desirable [45].”

Q2: From Line 382, the recommendation for AVS is mentioned. The Japanese JPAS study (PMID: 29092077), which determines the AVS recommendation based on CT finding and potassium values, may be helpful for discussion.

Answers to Q2: We discussed this subject and cited this publication:

“Researchers from JPAS (Japan Primary Aldosteronism Study) conducted a study investigating a combination of CT findings and serum potassium levels in the subtype diagnosis on AVS: 1591 patients with PA were included in this study and before AVS they underwent CT scan and serum potassium levels were evaluated. The results of this study have shown that patients with bilateral normal results on CT and normokalaemia could be treated pharmacologically because the probability of unilateral disease was low. Whereas patients with unilateral abnormalities on CT scan and hypokalemia were associated with a high probability of unilateral disease on AVS. Thus, AVS was strongly indicated for this group of patients [63].”

Q3: The pathogenesis of PA has been discussed since Line 79, focusing on somatic mutations. Molecular biological analysis focusing on somatic mutations is also in progress, including studies focusing on the transcriptome and methylome of KCNJ5 somatic mutations (e.g. PMID: 28747387) and on the production of hybrid steroid (e.g. PMID: 32206594, PMID: 31484828), which might be cited and mentioned as appropriate.

Answers to Q3: According to your suggestion, we have expanded the description of somatic mutations, including information about studies on the transcriptome and methylome of KCNJ5 mutated APAs, and the production of hybrid steroid characteristics for APAs with KCNJ5 mutation.

“In addition, the Japan Primary Aldosteronism Study (JPAS) conducted analysis for molecular characteristics of the KCNJ5 mutated APAs. This study demonstrated that APAs with somatic KCNJ5 mutation have global DNA hypomethylation and transcriptomic profiles accompanied by changes in specific genes such as Wnt signalling, cytokine, and inflammatory response pathways. Thus, the KCNJ5 mutated APAs constitute a specific subgroup among APAs [12]. Moreover, several studies investigating steroid profiling in PA diagnostics demonstrated hybrid steroid 18-hydroxycortisol (18OHF) and 18-oxocortisol (18oxoF) are characteristic for APAs with KCNJ5 mutations [16]. “

Reviewer Q4: Line 45: Line 45: What is meant by the description "FN"? Is it not FH (familial hyperaldosteronism)?

Answers to Q4: It should be FH (familial hyperaldosteronism). The mistake has been corrected.

Q5: Line 83: There is a description of ectopic aldosterone production. It is considered to be a very rare condition. Do authors have any references?

Answers to Q5: Reference PMID: 4124918 has been added.

Q6: Line 384:”AVA” should be “AVS”. Line 389: “ASV” should be “AVS”.

Answers to Q6: The mistake has been corrected.

Reviewer 2 Report

Comments and Suggestions for Authors

The minireview is of interest, although numerous reports have already been published on the subject.

comments

I would also highlight some new aspects of primary iperaldosteronism

-Please report the studies on selective thermoablation of APA

-Some reports from different laboratories describe adenomas of the adrenal that are non-steroidogenic or are surrounded by aldosterone-producing microadenomas and/or hyperplastic cells of the glomerulosa zone expressing CYP11B2. Some cases have been described of non-steroidogenic APA where the surrounding glomerulosa cells are the culprits.   In these cases, measurement of aldosterone in the adrenal veins might give contradictory results, in case thermoablation is used as selective therapy with excision of the adenoma. The ramifications of some APAs that also synthesize cortisol should also be considered

-Another important point to include is that spironolactone therapy acts not only on the aldosterone receptor but also on adrenal aldosterone synthesis by temporarily inhibiting aldosterone synthase. Insert work that has shown a cure of idiopathic hyperaldosteronism and in some cases adenoma after prolonged spironolactone therapy.

Author Response

Dear Reviewer,

We would like to extend my sincere gratitude for the valuable insights and constructive review regarding our manuscript. We are grateful for the time dedicated, and the valuable suggestions provided, which will undoubtedly contribute to further enhancing the quality of our work.

The answers to your comments can be found below.   Kind regards,   Natalia Ekman

Q1: Please report the studies on selective thermoablation of APA

Answers to Q1: As suggested, we have added the following description with relevant references.

“In addition, researchers are constantly investigating new less invasive methods of surgical treatment for patients with unilateral PA. One of these methods could be adrenal thermoablation as a treatment option for APAs in patients with PA. Thermoablation involves the use of temperature above 50° C to selectively remove the entire tumour without disturbing the surrounding healthy tissue. The advantage of thermoablation over adrenalectomy is the ability to spare the surrounding normal adrenal tissue and thus reduce the risk of adrenal insufficiency [81].                                                                                                                                                                                                                            The article “Aldosterone-producing Adenoma: Considerations on in Vitro Effects of Adrenal Thermoablation and its in Vivo Applications” suggests that thermal ablation can be effective in controlling blood pressure and reducing pharmacotherapy. The use of thermal ablation may provide a significant reduction in aldosterone levels and blood pressure in patients with APA [81].                                                                    In summary, thermal ablation is a promising treatment option for APAs, but some disadvantages and limitations should be considered such as lifelong adrenocortical insufficiency, appropriate technological design, advanced treatment planning, lack of a standardised protocol and relevant guidelines [82]. Therefore, further research and clinical trials of the long-term safety and effectiveness of thermoablation in APAs treatment, are needed.”

Q2: Some reports from different laboratories describe adenomas of the adrenal that are non-steroidogenic or are surrounded by aldosterone-producing microadenomas and/or hyperplastic cells of the glomerulosa zone expressing CYP11B2. Some cases have been described of non-steroidogenic APA where the surrounding glomerulosa cells are the culprits.   In these cases, measurement of aldosterone in the adrenal veins might give contradictory results, in case thermoablation is used as selective therapy with excision of the adenoma. The ramifications of some APAs that also synthesize cortisol should also be considered. 

Answers to Q2: Nonsteroidal adenomas surrounded by aldosterone-producing microadenomas and/or zona glomerular hyperplastic cells expressing CYP11B2 may pose a difficult clinical problem. Especially when thermal ablation is chosen as the treatment method for APA because in this case, selective removal of the adenoma does not cure primary aldosteronism. We described in our work the thermoablation of APAs, including the disadvantages and limitations of this method. We also mentioned cortisol co-secretion in the chapter about surgical treatment.

“It should also be noted that a significant proportion of patients may co-secrete cortisol, leading to contralateral adrenal cortisol suppression and a period of cortisol deficiency. This should be monitored and consideration given to transient hydrocortisone replacement [80].“

Q3: Another important point to include is that spironolactone therapy acts not only on the aldosterone receptor but also on adrenal aldosterone synthesis by temporarily inhibiting aldosterone synthase. Insert work that has shown a cure of idiopathic hyperaldosteronism and in some cases adenoma after prolonged spironolactone therapy.

Answers to Q3: We expanded the description associated with spironolactone, including inhibitory effects on steroidogenesis and production of aldosterone. We also added a reference about spontaneous remission of PA after long-term treatment of MRA.

“In addition, some studies have reported that MRAs such as spironolactone also have inhibitory effects on steroidogenesis and aldosterone production. This effect might contribute to the spontaneous remission of PA after long-term treatment with MRAs in some patients with PA. However, more research is needed to identify the specific patient characteristics, better understand the underlying mechanisms, and determine the optimal treatment duration for MRAs [88].”